# Sustained Release of Curcumin from Cur-LPs Loaded Adaptive Injectable Self-Healing Hydrogels

**DOI:** 10.3390/polym16243451

**Published:** 2024-12-10

**Authors:** Caixia Wu, Xiaoqun Ning, Qunfeng Liu, Xiaoyan Zhou, Huilong Guo

**Affiliations:** 1National Engineering Research Center for Healthcare Devices, Guangdong Provincial Key Laboratory of Medical Electronic Instruments and Materials, Institute of Biological and Medical Engineering, Guangdong Academy of Sciences, Guangzhou 510316, China; caixia52@163.com; 2Special Medical Service Center, Zhujiang Hospital, Southern Medical University, Guangzhou 510282, China; xiaoqun_ning@163.com; 3School of Automotive Engineering, Foshan Polytechnic, Foshan 528000, China; qunfengliu@163.com; 4Research Management Department, Guangdong Academy of Sciences, Guangzhou 510070, China

**Keywords:** injectable adhesive self-healing hydrogel, sustainable release of curcumin, antioxidant and anti-inflammatory, repair of biological tissue defect

## Abstract

Biological tissue defects are typically characterized by various shaped defects, and they are prone to inflammation and the excessive accumulation of reactive oxygen species. Therefore, it is still urgent to develop functional materials which can fully occupy and adhere to irregularly shaped defects by injection and promote the tissue repair process using antioxidant and anti-inflammatory mechanisms. Herein, in this work, phenylboronic acid modified oxidized hyaluronic acid (OHAPBA) was synthesized and dynamically crosslinked with catechol group modified glycol chitosan (GCHCA) and guar gum (GG) into a hydrogel loaded with curcumin liposomes (Cur-LPs) which were relatively uniformly distributed around 180 nm. The hydrogel possessed rapid gelation within 30 s, outstanding injectability and tissue-adaptive properties with self-healing properties, and the ability to adhere to biological tissues and adapt to tissue movement. Moreover, good biocompatibility and higher DPPH scavenging efficiency were illustrated in the hydrogel. And a more sustainable release of curcumin from Cur-LPs-loaded hydrogels, which could last for 10 days, was achieved to improve the bioavailability of curcumin. Finally, they might be injected to fully occupy and adhere to irregularly shaped defects and promote the tissue repair process by antioxidant mechanisms and the sustained release of curcumin for anti-inflammation. And the hydrogel would have potential application as candidates in tissue defect repair.

## 1. Introduction

Hydrogels have similar network structures to the biological soft tissue: they can absorb the exudates of biological tissue, maintain the tissue’s moistness, create a barrier against infection, and finally accelerate the repair of tissue defect. As a result, they have gradually been recognized as the most promising tissue repair materials [1,2,3,4]. Biological tissue defects are typically characterized by defects of various shapes and sizes; therefore, the first concern that should be taken into account is how to endow the hydrogels with the properties that enable it to be completely injected to occupy and adhere to the irregularly shaped defects and adapt to the tissue movement without being corrupted by external forces [5,6,7,8]. Dynamical bonds such as phenylboronester bonds, imine bonds, acylhydrazone bonds, coordination complex bonds, Diels–Alder reaction, and disulfide bonds are reversible covalent bonds that can lead to dynamic crosslinked networks. These can be either reversible covalently crosslinked networks or covalent adaptable networks [9,10] in the hydrogels, resulting in reversible performances such as shear-thinning (injectability) and self-healing properties that may enable the hydrogels to be injected, fully occupy the tissue defects with irregular shapes, and adapt to the movement of tissues and external forces in a completely integrated hydrogel state [11,12,13]. The biomimetic catechol adhesive group in mussels has been recently brought into hydrogels to enable them to be capable of adhering to the tissue to enhance their adaptability to tissue movement [5,14,15,16,17,18]. Therefore, through the molecular design of introducing catechol into dynamically crosslinked hydrogels, it is expected to obtain optimized tissue-adaptive injectable hydrogels that can be used for tissue repair, which will continue to expand the preparation of injectable hydrogels for tissue adaptation.

In addition, the repair of tissue defects is usually obstructed by increasing inflammation, which also causes a high expression of reactive oxygen species (ROS) [19,20,21,22,23]. The exacerbation of oxidative stress and inflammatory response caused by high levels of ROS will cause permanent and irreversible damage to biological tissue [24,25]. Therefore, biomaterials that are capable of scavenging ROS and reducing inflammation are urgently needed for the repair of tissue defects. Reduced catechol groups had been reported in our previous work to have excellent antioxidant performances [18]. And it is expected that the molecular design of introducing catechol into dynamically crosslinked hydrogels can not only improve the tissue adhesion performance but also achieve antioxidant performances to obtain optimized tissue-adaptive injectable hydrogels with scavenging ROS ability.

Curcumin is a natural compound with excellent anti-inflammatory properties, which has significant biological activity to reduce the inflammatory response of tissue defects. However, curcumin is a poorly soluble drug with a slow dissolution rate, resulting in poor absorption and low bioavailability in the body, which limits its clinical application [26,27,28]. It was reported that a drug delivery system which can increase water solubility with efficient loading and slow release can enhance the bioavailability of curcumin [29,30,31,32]. Liposomes have a phospholipid bilayer structure similar to cell membranes and exhibit good biocompatibility. They can encapsulate the oil-soluble drug molecule curcumin into the bilayer membrane of vesicles, forming a uniform and stable curcumin liposome solution. It can improve the bioavailability of curcumin. However, when liposome solutions are injected into tissues, their fluidity is often uncontrollable, leading to unclear targeting. Therefore, the target tissue’s sustained release of curcumin will be obtained, further improving the bioavailability of curcumin, through introducing Cur-LPs into tissue-adaptive injectable hydrogels constructed from catechol groups and dynamic crosslinking chemical bonds.

Herein, in this work, phenylboronic acid-modified oxidized hyaluronic acid (OHAPBA) was synthesized and dynamically crosslinked with the catechol group’s modified glycol chitosan (GCHCA) and guar gum (GG) into a hydrogel loaded with curcumin liposomes (Cur-LPs) (Figure 1). The introduction of catechol groups and the dynamic crosslinking network led to the hydrogel having excellent injectable performance as well as tissue-adaptive properties with self-healing capacities, tissue adhesion and adaptation to tissue movement. Moreover, the hydrogel possessed good biocompatibility and higher DPPH scavenging efficiency. And the bioavailability of curcumin might be improved due to the more sustainable release of curcumin from Cur-LPs loaded hydrogels. Finally, they might fully occupy and adhere to irregularly shaped defects by injection and promote the tissue repair process by antioxidant mechanisms and the sustained release of curcumin for anti-inflammation. And the hydrogel would have potential application prospects in tissue defect repair.

## 2. Experimental

### 2.1. Materials

Catechol-modified glycol chitosan (GCHCA) was prepared following the published methods in our previous literature [18]. Sodium hyaluronate without substitution (HA, 1000~1500 kDa), glycol chitosan (GC, M_w_ 100 kDa, DDA > 98%), and 3,4-dihydroxyphenyl acetic acid (HCA) were supplied by Shanghai yuanye Bio-Technology Co., Ltd. (Shanghai, China). 3-aminophenylboronic acid (APBA), Guar gum (GG, M_w_ 20 kDa), sodium (meta) periodate (NaIO_4_), and glycerol were obtained from Sigma-Aldrich. The molecular weight or other indicators of these raw materials are specified by the suppliers. 4-(4,6-Dimethoxy-1,3,5-triazin-2-yl)-4-methylmorpholinium chloride (DMTMM), 1-ethyl-3-[3-(dimethylamino) propyl]carbodiimide (EDC) and N-hydroxy succinimide (NHS) were obtained from Aladdin (Shanghai, China). Other reagents used as received were analytical or guaranteed reagents.

### 2.2. Preparation and Characterization of Catechol-Modified Glycol Chitosan (GCHCA)

Catechol-modified glycol chitosan (GCHCA) was prepared following the published methods in our previous literature [18]. Briefly, 2.4 g GC and 2 g HCA were completely dissolved into 300 mL MES buffer (0.1 M, pH 4.7) at 25 °C. Then, 2.1 g EDC and 1.2 g NHS in 20 mL MES buffer (0.1 M, pH 5.5) were dropped into the above solution under N_2_ atmosphere and stirred at 25 °C for 24 h. The mixtures were then dialyzed (MWCO 10,000) by acid pure water for 72 h to remove the unreacted HCA or salts and then continuously dialyzed (MWCO 10,000) by pure water for 10 h. Finally, the purified products were freeze-dried and stored at 2–8 °C before use. UV-vis spectroscopy and ^1^H NMR were performed to determine the structure of GCHCA, and the structural confirmation results could be seen in our previous literature [18].

### 2.3. Preparation and Characterization of Curcumin Liposomes (Cur-LPs)

Cur-LPs were prepared through the solvent evaporation method. In brief, 20 mg of curcumin, 120 mg of lecithin and 12 mg of cholesterol were dissolved in 10 mL of a mixed solution of chloroform and methanol (*v*/*v* = 1). And the solutions were evaporated at 40 °C to form a thin film, which was dried under vacuum for 24 h. Then, after suspending the thin film in PBS (0.01 M, pH 6) for 2 h, a probe-type ultrasonicator was used to prepare Cur-LPs. Centrifugation and a 0.22 μm filter membrane were used for the purification of liposomes. Finally, Cur-LPs were re-suspended in PBS solution at a certain mass concentration after freeze drying. The hydrodynamic size was determined by DLS (Zetasizer Nano ZS, Malvern, UK). And negative staining TEM (FEI Tecnai 12, Hillsboro, OR, USA) was used to investigate the morphology. Finally, curcumin contents in liposomes were determined by the standard curve of curcumin measured by UV-vis spectroscopy.
Curcumin content in liposomes C_c_ ug/mg = C_UV-vis_/C_Cur-LPs_

C_UV-vis_ is the curcumin concentration in Cur-LPs calculated by the standard curve of curcumin measured by UV-vis spectroscopy, and C_Cur-LPs_ is the mass concentration of Cur-LPs resuspension after freeze-drying.

And the encapsulation efficiency of curcumin in liposomes was calculated according to the below equation.
Encapsulation efficiency (%) = C_c-uv-vis_/C_c-theoretical_ ∗ 100%

C_c-uv-vis_ is the curcumin content in liposomes determined by UV-vis spectroscopy and C_c-theoretical_ is the theoretical value of the curcumin content in liposomes.

### 2.4. Preparation of Phenylboronic Acid Modified Oxidized Hyaluronic Acid Sodium Salt (OHAPBA) and Its Structural Characterization

OHA was prepared following the method published in our previous work [11]. First, 1 g OHA was dissolved in a mixture of 200 mL of pure water and DMF (the volume ratio of water to DMF is 3/2); then, 0.275 g of 3-aminophenylboronic acid (APBA) was added and stirred vigorously until it completely dissolved. After adding 0.388 g of DMTMM and continuously stirring for 24 h, the mixtures were dialyzed (MWCO 3500, Shanghai yuanye Bio-Technology Co., Ltd., Shanghai, China), freeze-dried, and stored at 2–8 °C before use. ^1^H NMR was used to confirm the structure of OHAPBA.

### 2.5. Preparation of the Hydrogels

First, 1 M NaOH was used to regulate the pH of GCHCA PBS solution at a concentration of 6% *w*/*v* to 7.5; after that, guar gum (GG) was dissolved into the GCHCA solutions until a homogeneous solution is formed. OHAAPBA (10% *w*/*v*) in a PBS solution was obtained simultaneously. Then, 200 μL of GCHCA/GG pre-gel solution and 200 μL of OHAPBA pre-gel solution were vigorously mixed at room temperature to form the hydrogel. Fluorescence spectroscopy (the excitation wavelength was in a fixation of 285 nm, and the emission spectrum was acquired at a wavelength of 280–500 nm) was performed to investigate the dynamic phenylboronester bond formation of the hydrogels.

### 2.6. Characterizations of the Hydrogels

#### 2.6.1. Swelling Ratios of the Hydrogels

The swelling ratios of the hydrogel were determined at 37 °C in PBS (0.1 M, pH 7.4) solution according to methods published in our previous work [18].
Swelling ratios (%) = W_swollen_/W_as-prepared_ ∗ 100%

W_swollen_ is the weight of the hydrogel in its swollen state, and W_as-prepared_ is the weight of the hydrogel as prepared.

#### 2.6.2. Rheological Tests

A rotational rheometer (KINEXOS Pro, Malvern) was used to test the hydrogels’ mechanical, shear-thinning and self-healing properties at 37 °C. The amplitude sweep test over different strains (0.1% to 800%) was used to analyze the storage modulus (G′) changes within the linear viscoelastic region. The frequency sweep (0.1 1/s to 100 1/s) was performed to investigate the shear-thinning property, and the G′ and G″ test at alternate step strain cycles of 1% and 500% was carried out to evaluate the self-healing behavior of the hydrogel.

#### 2.6.3. Cytotoxicity

CCK-8 assay was performed to assess the cytotoxic property of the hydrogels loaded with or without Cur-LPs toward mouse 3T3 fibroblast following the methods published previously [18,33,34], which was performed in 24-well chambers (Corning Costar, Tewksbury, MA, USA) equipped with PET track-etched membranes (pore size = 0.8 μm) on the bottom of the cell culture insert. Then, 100 μL of the sterilized hydrogels with or without Cur-LPs (the sterilized hydrogels were prepared in an ultraclean table (BBS-DDC, BIOBASE, Jinan, China) by means of ultraviolet sterilization with 3 mW/cm^2^ for 30 min, preparation of pre-gel solution from the sterilization solution, and sterilization of the pre-gel solution through 0.22 μm filter membrane) was introduced to the upper insert separately and kept at 37 °C. The 3T3 cell was seeded in the lower chambers. After 48 h of incubation, a CCK-8 assay was performed to investigate the cytotoxic properties. Five multiple holes were set up for each sample.
Cell viability (%) = [A_sample_ − A_blank_]/[A_control_ − A_blank_] ∗ 100%

A_sample_: absorbance of cells and CCK-8 solution after co-incubation with hydrogels;

A_blank_: absorbance of medium and CCK-8 solution without cells;

A_control_: absorbance of medium and CCK-8 solution with cells.

#### 2.6.4. Hemolysis Evaluation of Hydrogels

The fresh mouse blood protocol was approved by the Animal Ethics Committee of Institute of Biological and Medical Engineering, Guangdong Academy of Sciences (protocol code K2024-02-154-158 and date of approval 22 July 2024). Then, the sterile PBS was used to dilute the fresh mouse blood 16-fold. Afterwards, 100 mg of hydrogels in a 24-well plate was incubated with 500 μL of erythrocyte for each well at 37 °C for 1 h with a shaking speed of 100 rpm. Centrifugation was performed at 3500 rpm for 5 min to separate the non-hemolyzed erythrocyte. The microplate reader at 545 nm was used to measure the absorbance of supernatants. Hemolysis ratios were calculated according to the below eq.
Hemolysis ratio (%) = [(OD_sample_ − OD_PBS_)/(OD_water_ − OD_PBS_)] × 100%

#### 2.6.5. DPPH Scavenging Efficiency of Hydrogels

The DPPH scavenging assay that may evaluate antioxidant performance was investigated. Samples with different concentrations were mixed with 3 mL of DPPH solution and allowed to react in the dark for 30 min. Then, the full wavenumber scanning curves were recorded, and the absorbance values at 517 were measured. The DPPH scavenging ratios were calculated by the following formula: DPPH scavenging ratio % = (A_b_ − A_h_)/A_b_ ∗ 100%

Here, A_b_ and A_h_ were the absorption of the blank (DPPH + ethanol) and the absorption of the hydrogel (DPPH + ethanol + hydrogel), respectively. Hydrogel 0 mg is defined as the control group, and the DPPH scavenging ratio % of the control group is 0%.

#### 2.6.6. In Vitro Release of Curcumin from Cur-LPs Loaded Hydrogel

The same amounts of Cur LPs in 1 mL PBS or hydrogel were encapsulated in the dialysis bags (Mw 3500), which were immersed in PBS (0.1 M, pH 7.4) with 1% Tween-80. After being incubated in a shaker at 37 °C under 100 rpm for predetermined periods, 1 mL of supernatant was withdrawn for UV-Vis analyses and replaced with 1 mL of PBS. All in vitro release studies were carried out in triplicate, and their average values were recorded. The cumulative curcumin release as a function of time was analyzed.

### 2.7. Statistical Analysis

Every group with five parallel data was presented as mean ± SD. ANOVA and the T test were used to measure statistical differences in data (* *p* < 0.05 significant, ** *p* < 0.01 very significant).

## 3. Results and Discussion

From Figure 2a,b, it could be seen that the average particle size is 180 nm with a relatively narrow distribution of hydrodynamic size (PDI = 0.45) in the DLS results, and relative uniformly distributed nanoparticles are also shown in the TEM image of Cur-LPs. Figure 2c also shows a similar spectrum of UV-Vis absorption in curcumin and Cur-LPs. And the curcumin content in liposomes, calculated by the standard curve of curcumin measured by UV-vis spectroscopy (Figure 2d), was 97 μg/mg with 64% encapsulation efficiency [35]. ^1^H NMR spectroscopy was performed to confirm the chemical structure of OHAPBA, as shown in Figure 3. The grafting rate of the phenylboronic acid group was 6.8 ± 0.2% through calculating the integral area at 1.91 ppm corresponding to—CH_3_ and 7.01 ppm corresponding to the hydrogen on the benzene ring. And the dynamically crosslinked network formed by a dynamic phenylboronester bond had been proved by Figure 3b, from which a phenomenon of fluorescence quenching [36,37] could be seen in OHAPBA after being mixed with GCHCA and GG, confirming the formation of a dynamic phenylboronic ester bond between the catechol group on GCHCA or hydroxyl group on GG and phenylboronic acid in OHAPBA.

The inversion method was carried out to investigate the gelation process of the hydrogel, as shown in Figure 4a. The mixed pre-gel solution turned into a colloid and lost its fluidity in 25 s when inverted, confirming the quick gelation of the hydrogel. And the hydrogel crosslinked through dynamic chemical bonds could be injected through a needle of 26G easily without blocking the needle, as shown in Figure 4b. This indicated that the injectable hydrogel is conducive to embedding biomaterials and can be injected to completely fill irregular tissue defects [38,39]. And the outstanding injectability was owing to the shear-thinning properties of the hydrogel, which are illustrated in Figure 4c. The complex viscosity reduced significantly as the shear rate increased, which demonstrated a typical shear thinning behavior of injectable hydrogel [40]. In addition, swelling performance, which endows the hydrogel with the ability to absorb exudates and maintain a humid environment when used as implanted materials for repairing tissue defects, should be investigated before use [5]. A higher swelling ratio might impose physical compression or impair the toughness around tissue defects, which was detrimental to the repair of tissue defects [41,42]. Figure 4d shows the swelling rate of the as-prepared hydrogels, which demonstrated the swelling ratios in an appropriate value lower than 220%. This indicated that the hydrogel with an appropriate swelling rate can be very advantageously used as embedding biomaterials for tissue repair. Moreover, as shown in Figure 4c,e, comparing the storage moduli G″ and complex viscosity of OHAPBA/GCHCA hydrogel and OHAPBA/GCHCA/GG hydrogel, the mechanical performances was enhanced by the addition of GG, which might be due to the role of GG itself as a thickener and the formation of more hydrogen bonds between the molecular chain of the hydrogel. And there was no significant effect on the mechanical properties with the addition of Cur-LPs into the hydrogel.

As shown in Figure 5a, when the two halves of the cracked hydrogels were contacted immediately, they would be healed to an integrated hydrogel, which was capable of withstanding certain tension without damaging. And from the G′ and G″ test at alternative strains in Figure 5b, it could be seen that the hydrogel will yield at a higher strain of 500%, whereas the mechanical property was restored quickly and was accompanied by G′ and G″ recovering to the original state when resetting to a low strain of 1%. Furthermore, the stable self-healing behavior illustrated in Figure 5b after four alternative strain cycles was attributed to the hydrogel network crosslinked by dynamic bonds. And the potential dissociative reversion mechanisms [43] of covalent adaptable networks that were dynamic crosslinked by dynamic phenylboronic esters bonds are illustrated schematically in Figure 5c, from which the self-healing property of the hydrogel can be understood more clearly. In addition to self-healing performance, tissue adhesion is also critical to the tissue adaptability of hydrogel, which is shown in Figure 5d. The Cur-LPs-loaded OHAPBA/GCHCA/GG hydrogel adhered very well onto the finger skin surface and adapted to different finger movements. The excellent adhesion performance might come from the interaction between the hydroxyl or catechol group in the hydrogel and typical group on the surface of the skin tissue (Figure 5d).

Biocompatibilities of the hydrogel, which were assessed by cell compatibility and blood compatibility, need to be investigated before being applied in biomedicine. A CCK-8 [11,18] assay was carried out to evaluate the cell compatibility, as shown in Figure 6a. Almost 100% of mouse 3T3 fibroblasts survived after co-incubation with OHAPBA/GCHCA/GG hydrogel with different concentrations of Cur-LPs or without Cur-LPs for forty-eight hours. This indicated the lack of cytotoxicity and excellent cytocompatibility of the hydrogel. And the hemolytic ratio experiment, as shown in Figure 6b, was performed to analyze the hemocompatibility. The erythrocytes, co-incubated with OHAPBA/GCHCA/GG hydrogel with different concentrations of Cur-LPs or without Cur-LPs, demonstrated similar morphologies with the group co-incubated with PBS. And there were no significant differences in the hemolysis ratios, which were obviously below the critical limit allowed by 5% [5], among the groups co-incubated with PBS and with OHAPBA/GCHCA/GG hydrogel with different concentration of Cur-LPs or without Cur-LPs. This indicated the excellent hemocompatibility illustrated in the hydrogels. Furthermore, DPPH scavenging assay may be used to evaluate antioxidant performance, which was also crucial for the hydrogels used in biomedicine for tissue repair, as shown in Figure 6c. The DPPH scavenging efficiency of the OHAPBA/GCHCA/GG hydrogels with or without Cur-LPS was higher than 35% even at lower co-incubation concentrations of 0.5 mg/mL. The IC50 values calculated according to the linear relationship between DPPH scavenging efficiency and the concentrations of the hydrogels were 0.81 mg/mL for OHAPBA/GCHCA hydrogel and 0.79 mg/mL for OHAPBA/GCHCA/GG hydrogel, respectively. And the DPPH scavenging ratio, which increased with the increment of the hydrogel concentration, achieved almost 100% at 4 mg/mL of the co-incubation concentration, which was attributed to the addition of the reduced catechol group [44,45]. The good cytotocompatibility, hemotocompatibility and higher DPPH scavenging efficiency inspired us that the Cur-LPs loaded OHAPBA/GCHCA/GG hydrogel may be used safely in biomedicine.

In order to further assess the application performance of Cur-LPs loaded OHAPBA/GCHCA/GG hydrogel in biomedicine, in vitro curcumin (an anti-inflammatory agent) release from the liposome and the hydrogel was analyzed, as shown in Figure 6d. There was more sustained curcumin release in the Cur-LPs-loaded OHAPBA/GCHCA/GG hydrogel compared with that release in liposomes. The more sustained curcumin release property could extend the action time, reduce the frequency of administration, and finally improve the bioavailability of curcumin.

From all of the above, it could be seen that curcumin liposomes were loaded into adaptive injectable self-healing hydrogels formed by a dynamic crosslinked network. The increased water solubility with efficient loading and slow release performances of curcumin indirectly proved the enhancement of bioavailability. However, the direct evidence of bioavailability is not provided in this work, which is also the limitation of this work. In our future work, the direct evidence of the improvement of bioavailability and the in vivo application for tissue repair, e.g., wound healing, bone regeneration or periodontitis, will be investigated.

## 4. Conclusions

In summary, hydrogels with outstanding injectability and tissue-adaptive properties including self-healing, adhesion to biological tissues, and adaptation to tissue movement were achieved due to the introduction of a catechol group and dynamic networks crosslinked by phenylboronic acid modified oxidized hyaluronic acid (OHAPBA) and catechol group-modified glycol chitosan (GCHCA) or guar gum (GG). And the hydrogel illustrated good biocompatibility and higher DPPH scavenging efficiency. Moreover, a more sustainable release of curcumin from Cur-LPs loaded hydrogels was achieved, which might improve the bioavailability of curcumin to enhance the anti-inflammation performance of the hydrogel. Finally, reasonable assumptions can be derived from the above that they might be injected to fully occupy and adhere to irregularly shaped defects and promote the tissue repair process by the antioxidant and sustained release of curcumin for anti-inflammation. And the hydrogel would have potential applications as candidates in tissue defect repair.

## Figures and Tables

**Figure 1 polymers-16-03451-f001:**
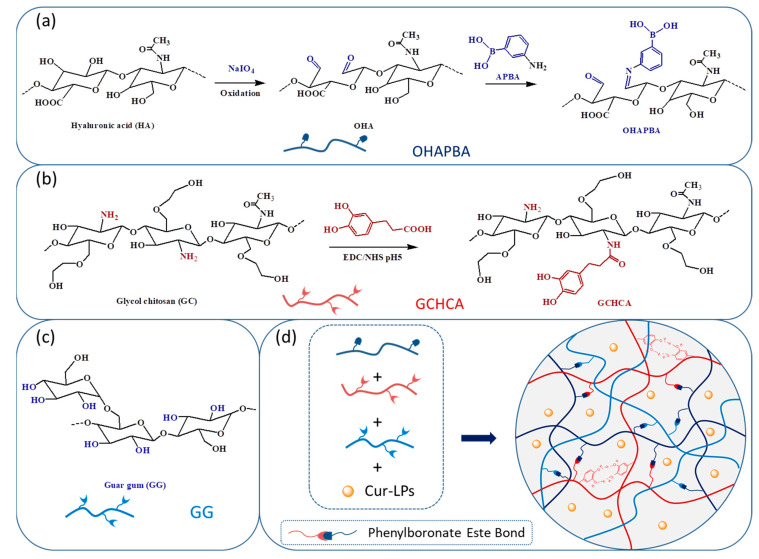
(**a**) Preparation of OHAPBA; (**b**) preparation of GCHCA; (**c**) structure diagram of GG; (**d**) preparation of the hydrogel and schematic diagram of the dynamic networks.

**Figure 2 polymers-16-03451-f002:**
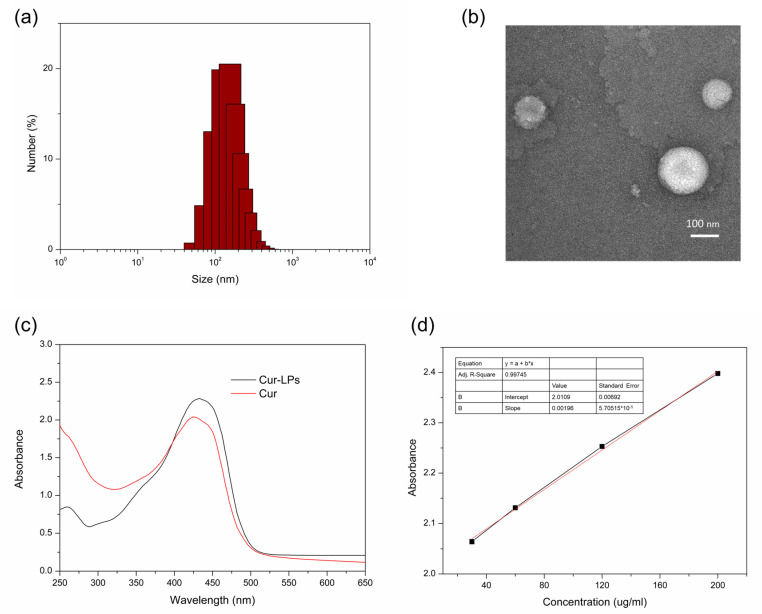
(**a**) DLS; (**b**) TEM; (**c**) UV-vis spectrum of Cur-LPs; (**d**) standard curve of curcumin.

**Figure 3 polymers-16-03451-f003:**
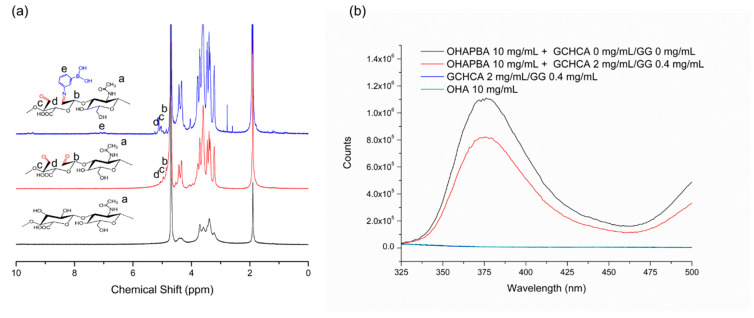
(**a**) ^1^H NMR spectroscopy of OHAPBA, OHA and HA; (**b**) fluorescence spectroscopy of OHAPBA with GCHCA and GG.

**Figure 4 polymers-16-03451-f004:**
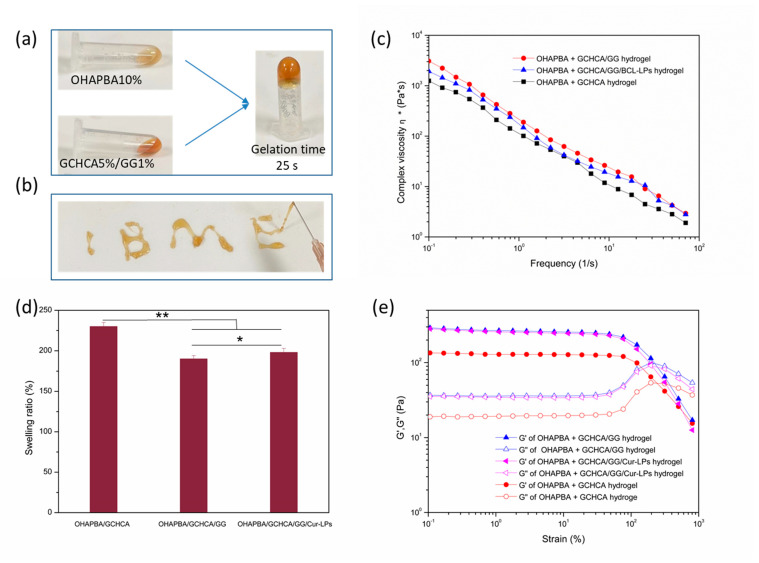
(**a**) Gelation process by inverted method; (**b**) injectable performance; (**c**) shear-thinning behavior; (**d**) swelling ratios; (**e**) G′ and G″ changes versus strain. (* *p* < 0.05 significant, ** *p* < 0.01 very significant).

**Figure 5 polymers-16-03451-f005:**
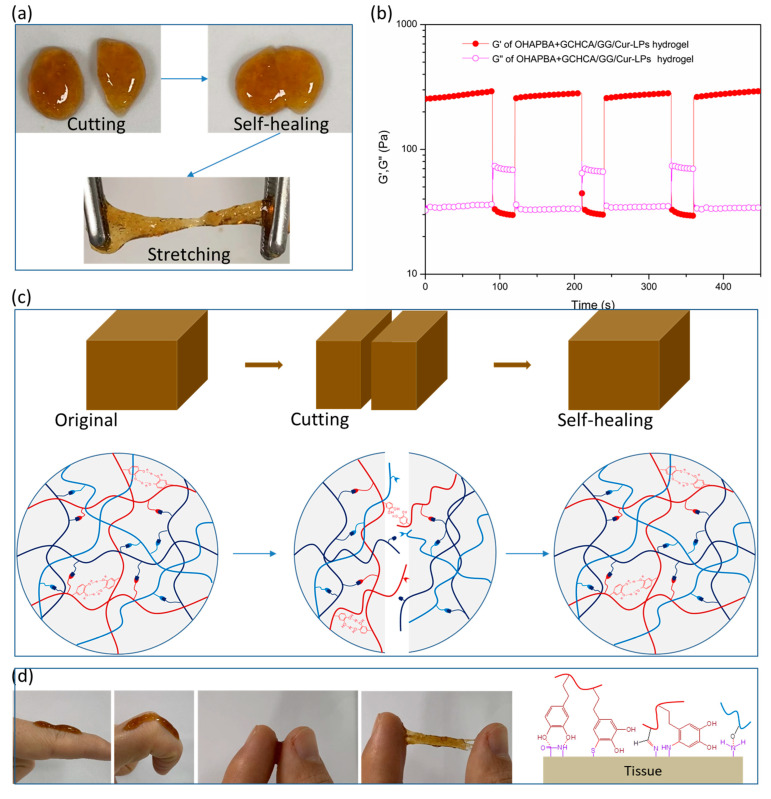
(**a**) Self-healing property illustrated by two-cracked hydrogel pieces; (**b**) G′ and G″ recovery after four cycles of alternating strain; (**c**) schematic diagram of the self-healing mechanism; (**d**) adhesive property, adaptability on finger skin of the hydrogel and typical reaction of hydrogel adhesion on tissue.

**Figure 6 polymers-16-03451-f006:**
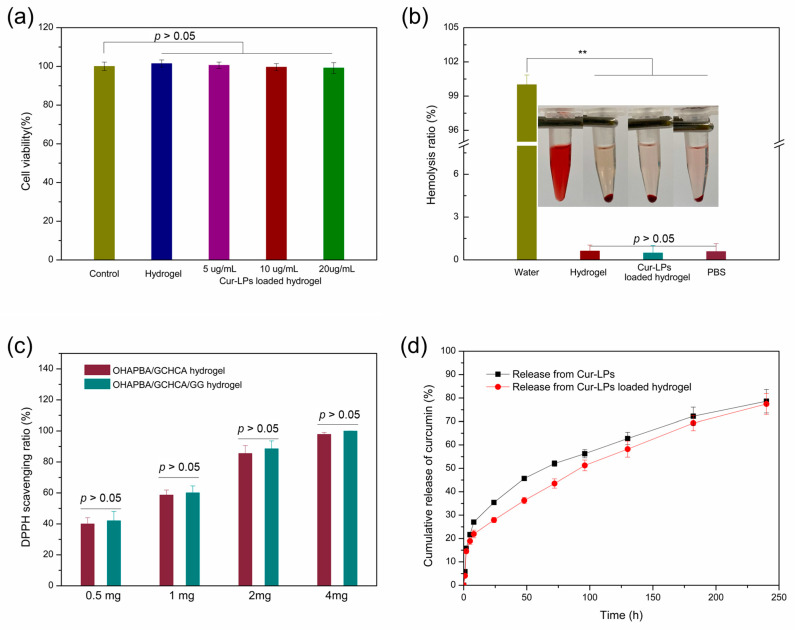
(**a**) Cell viability calculated from CCK-8 results; (**b**) hemolytic property; (**c**) DPPH scavenging ratio; (**d**) release of curcumin from Cur-LPs-loaded hydrogel. (** *p* < 0.01 very significant).

## Data Availability

The original contributions presented in this study are included in the article. Further inquiries can be directed to the corresponding author.

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
