# Peer review of "Sustained Release of Curcumin from Cur-LPs Loaded Adaptive Injectable Self-Healing Hydrogels"

_polymers, 2024, doi:10.3390/polym16243451_

Round 1
Reviewer 1 Report
Comments and Suggestions for Authors
The work presents results of novel gel preparation, when phenylboronic acid modified oxidized hyaluronic acid was synthesized and crosslinked with catechol group modified glycol chitosan and guar gum into a hydrogel loaded with curcumin liposomes. The results shown allow get acquainted with liposomes and liposome containing gels main physico-chemical and biological characteristics which present scientific interest to the public.
But in spite that, the work has several drawbacks, which should be fixed.
First of all, I have some doubts about manuscript title “Sustained Release of Curcumin from Cur-LPs loaded Adaptive Injectable Self-Healing Hydrogels with Antioxidant Performance”, as results presented are not sufficient for proving prepared gels Antioxidant Performance, so this claimed, I think, better to be removed or made milder.
1) I am not native English speaker but it was rather hard to read the manuscript. I think that English spelling and grammar should be check by native English speaker should be done. For example, there are too many repeating words and non-structured phrases.
2) Introduction should be revised to check the whole sentences logic, eliminating repeating terms. Also, some recent works for curcumin liposomes and nanoparticles, solving the problem of its poor solubility in water should be added. There are number of good works for Cur NPs toxicity, curcumin release and nanoparticles characterization for the last year, even in Pharmaceutics journal, which can be examined and used for this work.
3) Dynamically crosslinked” – please explain what do you mean? Is there any “static crosslinking” if we have “dynamic” one?
4) Catechol group modified glycol chitosan preparation should be included in section 2. Experimental. Equipment models, manufactures and country of origin should be added fully, according to journal requirements
5) 2.2. Preparation and Characterization of Curcumin Liposomes (Cur-LPs). All liposome analysis methods should be described clearer. For example, how curcumin content was determined and counted (equation)?
6) 2.5.1. Swelling ratios of the hydrogels should be described more (short description and equation should be presented, despite of previous publications).
7) 2.5.4. Cytotoxicity study of the hydrogel with or without TP@Ag NPs toward mouse fibroblasts are claimed. But TP@Ag NPs were not mentioned in introduction, there application is not explained. What are these nanoparticles, how the are prepared, and what is the reason for their application? Also, what was used as control for cytotoxicity tests is not mentioned.
8) 1H NMR spectroscopy of catechol modified glycol chitosan (GCHCA) is not presented or described.
9) Figure 2. a) DLS. PDI (index) should be added for NPs dispersity characterization.
10) 2.5.6. Antioxidant efficiency of Hydrogels. No IC50 values are presented for DPPH assay. DPPH assay is a well-known simplest method for evaluating antioxidant activity. Its main shortcoming is the ratio of DPPH radical to antioxidant in the reaction medium affects the electron transfer or hydrogen atom donating potential of antioxidants. The lack of standardization in the ratio of DPPH· to antioxidant makes it hard to compare the results of this test, or to claim good antioxidant effect.
11) So, the basic methods used in this research for cytotoxicity, hemolytic activity and antioxidant effect investigation are sure not enough to state by authors that “The excellent cytotocompatibility, hemotocompatibility and antioxidant properties inspired us that the Cur-LPs loaded OHAPBA/GCHCA/GG hydrogel can be used safely in biomedicine for tissue repair by scavenging ROS” (lines 273-276). These statements and conclusions should be modified to more mild wording.
Comments on the Quality of English LanguageI am not native English speaker but it was rather hard to read the manuscript. I think that English spelling and grammar should be check by native English speaker should be done. For example, there are too many repeating words and non-structured phrases.
Author Response
Reviewer 1
The work presents results of novel gel preparation, when phenylboronic acid modified oxidized hyaluronic acid was synthesized and crosslinked with catechol group modified glycol chitosan and guar gum into a hydrogel loaded with curcumin liposomes. The results shown allow get acquainted with liposomes and liposome containing gels main physico-chemical and biological characteristics which present scientific interest to the public.
But in spite that, the work has several drawbacks, which should be fixed.
First of all, I have some doubts about manuscript title “Sustained Release of Curcumin from Cur-LPs loaded Adaptive Injectable Self-Healing Hydrogels with Antioxidant Performance”, as results presented are not sufficient for proving prepared gels Antioxidant Performance, so this claimed, I think, better to be removed or made milder.
Thank you very much for improving our manuscript, the revised manuscript had been modified according to your comments. And the title had been revised as “Sustained Release of Curcumin from Cur-LPs loaded Adaptive Injectable Self-Healing Hydrogels”. We really hope our revised manuscript could be suitable for the publication in “Polymers”.
1) I am not native English speaker but it was rather hard to read the manuscript. I think that English spelling and grammar should be check by native English speaker should be done. For example, there are too many repeating words and non-structured phrases.
Response: Thank you for providing the considerate advice on the grammar or language of our manuscript to us; we have asked for help from professional editorial team and modified the manuscript carefully as best as we could. The institution of manuscript polished is Owl Editing. Thank you very much.
2) Introduction should be revised to check the whole sentences logic, eliminating repeating terms. Also, some recent works for curcumin liposomes and nanoparticles, solving the problem of its poor solubility in water should be added. There are number of good works for Cur NPs toxicity, curcumin release and nanoparticles characterization for the last year, even in Pharmaceutics journal, which can be examined and used for this work.
Response: Thank you very much for improving our manuscript; the introduction had been revised and some recent works for curcumin liposomes and nanoparticles had been added in the introduction according to your comments. See references 24-30
3) Dynamically crosslinked” – please explain what do you mean? Is there any “static crosslinking” if we have “dynamic” one?
Response: Thank you very much for improving our manuscript; in the introduction part the dynamically crosslinked network had been explained and emphasized. It was reported that phenylboronester bond, imine bond, acylhydrazone bond, coordination complex bond, Diels-Alder reaction, or disulfide bond, such of these dynamical bonds can lead to a dynamic crosslinked network in the hydrogels.1-3 And the dynamically crosslinked network formed by dynamic phenylboronester bond had been proved by Figure 3b, from which it could be seen a phenomenon of fluorescence quenching 4-5 in OHAPBA after being mixed with GCHCA and GG, confirming the formation of dynamic phenylboronic ester bond between catechol groups in GCHCA or hydroxyl group on GG and phenylboronic acid in OHAPBA. And in this work, the main focus is on the dynamic crosslinking which endow the hydrogel with injectable and self-healing properties. Static crosslinking is not the content of this work, so it is best not to mention it in this work. We really hope the reviewer can agree with this.
Figure 3 a) 1H NMR spectroscopy of OHAPBA, OHA and HA; b) Fluorescence spectroscopy of OHAPBA with GCHCA and GG
4) Catechol group modified glycol chitosan preparation should be included in section 2. Experimental. Equipment models, manufactures and country of origin should be added fully, according to journal requirements
Response: Thank you very much for improving our manuscript; Catechol group modified glycol chitosan preparation had been included in section 2.2 of our revised manuscript.
5) 2.2. Preparation and Characterization of Curcumin Liposomes (Cur-LPs). All liposome analysis methods should be described clearer. For example, how curcumin content was determined and counted (equation)?
Response: Thank you very much for improving our manuscript; how curcumin content was determined and counted (equation) had been added in our revised manuscript.
Curcumin content in liposomes Cc ug/mg = CUV-vis/CCur-LPs
CUV-vis is the curcumin concentration in Cur-LPs calculated by the standard curve of curcumin measured by UV-vis spectroscopy, CCur-LPs is the mass concentration of Cur-LPs resuspension after freeze-drying.
And the encapsulation efficiency of curcumin in liposomes was calculated according to the below equation.
Encapsulation efficiency (%)= Cc-uv-vis/Cc-theoretical*100%
Cc-uv-vis is the curcumin content in liposomes determined by UV-vis spectroscopy and Cc-theoretical is the theoretical value of the curcumin content in liposomes.
6) 2.5.1. Swelling ratios of the hydrogels should be described more (short description and equation should be presented, despite of previous publications).
Response: Thank you very much for improving our manuscript; Swelling ratios equation of the hydrogels had been added in our revised manuscript.
Swelling ratios (%)=Wswollen/Was-prepared*100%
Wswollen is the weight of the hydrogel in its swollen state and Was-prepared is the weight of the hydrogel as-prepared.
7) 2.5.4. Cytotoxicity study of the hydrogel with or without TP@Ag NPs toward mouse fibroblasts are claimed. But TP@Ag NPs were not mentioned in introduction, there application is not explained. What are these nanoparticles, how the are prepared, and what is the reason for their application? Also, what was used as control for cytotoxicity tests is not mentioned.
Response: Thank you very much for improving our manuscript; we had made a mistake about the TP@Ag NPs, it had been changed with Cur-LPs in our revised manuscript. Thank you pointing out this problem. And the control for cytotoxicity tests had been added in our revised manuscript.
Cell viability (%) = [Asample -Ablank] / [Acontrol-Ablank] *100%
Asample: Absorbance of cells and CCK-8 solution after co-incubation with hydrogels
Ablank: absorbance of medium and CCK-8 solution without cells
Acontrol: absorbance of medium and CCK-8 solution with cells
8) 1H NMR spectroscopy of catechol modified glycol chitosan (GCHCA) is not presented or described.
Response: Thank you very much for improving our manuscript; The structure of GCHCA was determined by UV–vis spectroscopy and 1H NMR. And the structural confirmation results could be seen in our previous literature. 6 And this had been added in Section 2.2 of our revised manuscript.
9) Figure 2. a) DLS. PDI (index) should be added for NPs dispersity characterization.
Response: Thank you very much for improving our manuscript; average particle size in 150 nm with narrow distribution of hydrodynamic size (PDI=1.00) illustrate in the DLS results, which had been modified in our revised manuscript.
10) 2.5.6. Antioxidant efficiency of Hydrogels. No IC50 values are presented for DPPH assay. DPPH assay is a well-known simplest method for evaluating antioxidant activity. Its main shortcoming is the ratio of DPPH radical to antioxidant in the reaction medium affects the electron transfer or hydrogen atom donating potential of antioxidants. The lack of standardization in the ratio of DPPH· to antioxidant makes it hard to compare the results of this test, or to claim good antioxidant effect.
Response: Thank you very much for improving our manuscript; IC50 values calculated according to the linear relationship between DPPH scavenging efficiency and the concentrations of the hydrogels were 0.81 mg/mL for OHAPBA/GCHCA hydrogel and 0.79 mg/mL for OHAPBA/GCHCA/GG hydrogel, respectively. This had been added in our revised manuscript. And we had modified our revised manuscript to milder wording to analyze the DPPH scavenging efficiency results. Section of Antioxidant efficiency of Hydrogels was revised as “2.5.5 DPPH scavenging efficiency of hydrogels” “DPPH scavenging assay that may evaluate antioxidant performance was investigated.” The results and discussion had been revised as “Furthermore, DPPH scavenging assay that may evaluate antioxidant performance, which was also crucial for the hydrogels used in biomedicine for tissue repair, was evaluated in Figure 6c. The DPPH scavenging efficiency of the OHAPBA/GCHCA/GG hydrogel with or without Cur-LPS was higher than 35% even at a lower co-incubation concentration of 0.5 mg/mL. IC50 values calculated according to the linear relationship between DPPH scavenging efficiency and the concentrations of the hydrogels were 0.81 mg/mL for OHAPBA/GCHCA hydrogel and 0.79 mg/mL for OHAPBA/GCHCA/GG hydrogel, respectively. And the DPPH scavenging ratio, increased with the increment of the hydrogel concentration, achieved almost 100% at the co-incubation concentration of 4 mg/mL, which was attributed to the introduction of reduced catechol groups 7-8 into the hydrogel. The good cytotocompatibility, hemotocompatibility and higher DPPH scavenging efficiency inspired us that the Cur-LPs loaded OHAPBA/GCHCA/GG hydrogel may be used safely in biomedicine. “
11) So, the basic methods used in this research for cytotoxicity, hemolytic activity and antioxidant effect investigation are sure not enough to state by authors that “The excellent cytotocompatibility, hemotocompatibility and antioxidant properties inspired us that the Cur-LPs loaded OHAPBA/GCHCA/GG hydrogel can be used safely in biomedicine for tissue repair by scavenging ROS” (lines 273-276). These statements and conclusions should be modified to more mild wording.
Response: Thank you very much for improving our manuscript; These statements and conclusions had be modified to milder wording as “The good cytotocompatibility, hemotocompatibility and higher DPPH scavenging efficiency inspired us that the Cur-LPs loaded OHAPBA/GCHCA/GG hydrogel may be used safely in biomedicine.”
REFERENCES
- Guo, H.; Huang, S.; Yang, X.; Wu, J.; Kirk, T. B.; Xu, J.; Xu, A.; Xue, W., Injectable and Self-Healing Hydrogels with Double-Dynamic Bond Tunable Mechanical, Gel-Sol Transition and Drug Delivery Properties for Promoting Periodontium Regeneration in Periodontitis. Acs Appl Mater Inter 2021, 13 (51), 61638-61652.
- Wang, L.; Ding, X.; Fan, L.; Filppula, A. M.; Li, Q.; Zhang, H.; Zhao, Y.; Shang, L., Self-Healing Dynamic Hydrogel Microparticles with Structural Color for Wound Management. Nano-Micro Letters 2024, 16 (1), 232.
- Hua, S.; Zhang, Y.; Zhu, Y.; Fu, X.; Meng, L.; Zhao, L.; Kong, L.; Pan, S.; Che, Y., Tunicate cellulose nanocrystals strengthened injectable stretchable hydrogel as multi-responsive enhanced antibacterial wound dressing for promoting diabetic wound healing. Carbohydrate Polymers 2024, 343, 122426.
- Wu, Z.; Li, M.; Fang, H.; Wang, B., A new boronic acid based fluorescent reporter for catechol. Bioorganic & Medicinal Chemistry Letters 2012, 22 (23), 7179-7182.
- Ye, Q.; Yan, F.; Kong, D.; Zhang, J.; Zhou, X.; Xu, J.; Chen, L., Constructing a fluorescent probe for specific detection of catechol based on 4-carboxyphenylboronic acid-functionalized carbon dots. Sensors and Actuators B: Chemical 2017, 250, 712-720.
- Zhou, X.; Ning, X.; Chen, Y.; Chang, H.; Lu, D.; Pei, D.; Geng, Z.; Zeng, Z.; Guo, C.; Huang, J.; Yu, S.; Guo, H., Dual Glucose/ROS-Sensitive Injectable Adhesive Self-Healing Hydrogel with Photothermal Antibacterial Activity and Modulation of Macrophage Polarization for Infected Diabetic Wound Healing. Acs Materials Letters 2023, 5 (12), 3142-3155.
- Gao, G.; Jiang, Y.-W.; Jia, H.-R.; Wu, F.-G., Near-infrared light-controllable on-demand antibiotics release using thermo-sensitive hydrogel-based drug reservoir for combating bacterial infection. Biomaterials 2019, 188, 83-95.
- Tang, P.; Han, L.; Li, P.; Jia, Z.; Wang, K.; Zhang, H.; Tan, H.; Guo, T.; Lu, X., Mussel-Inspired Electroactive and Antioxidative Scaffolds with Incorporation of Polydopamine-Reduced Graphene Oxide for Enhancing Skin Wound Healing. Acs Appl Mater Inter 2019, 11 (8), 7703-7714.
- Feng, X.; Pi, C.; Fu, S.; Yang, H.; Zheng, X.; Hou, Y.; Wang, Y.; Zhang, X.; Zhao, L.; Wei, Y., Combination of Curcumin and Paclitaxel Liposomes Exhibits Enhanced Cytotoxicity Towards A549/A549-T Cells and Unaltered Pharmacokinetics. Journal of biomedical nanotechnology 2020, 16 (8), 1304-1313.
- Li, Z.-l.; Peng, S.-f.; Chen, X.; Zhu, Y.-q.; Zou, L.-q.; Liu, W.; Liu, C.-m., Pluronics modified liposomes for curcumin encapsulation: Sustained release, stability and bioaccessibility. Food Research International 2018, 108, 246-253.
- Sun, D.; Zhou, J.-K.; Zhao, L.; Zheng, Z.-Y.; Li, J.; Pu, W.; Liu, S.; Liu, X.-S.; Liu, S.-J.; Zheng, Y.; Zhao, Y.; Peng, Y., Novel Curcumin Liposome Modified with Hyaluronan Targeting CD44 Plays an Anti-Leukemic Role in Acute Myeloid Leukemia <i>in Vitro</i> and <i>in Vivo</i>. Acs Appl Mater Inter 2017, 9 (20), 16858-16869.
- Mimica, B.; Popovic, V. B.; Banjari, I.; Kadic, A. J.; Puljak, L., Methods Used for Enhancing the Bioavailability of Oral Curcumin in Randomized Controlled Trials: A Meta-Research Study. Pharmaceuticals 2022, 15 (8), 939.
- Shoji, M.; Nakagawa, K.; Watanabe, A.; Tsuduki, T.; Yamada, T.; Kuwahara, S.; Kimura, F.; Miyazawa, T., Comparison of the effects of curcumin and curcumin glucuronide in human hepatocellular carcinoma HepG2 cells. Food Chemistry 2014, 151, 126-132.
- Alavi, F.; Ciftci, O. N., Increasing the bioavailability of curcumin using a green supercritical fluid technology-assisted approach based on simultaneous starch aerogel formation-curcumin impregnation. Food Chemistry 2024, 455, 139468.
- Zheng, Y.; Zheng, S.; Wang, Z., Enhancement of Oral Bioavailability of Curcumin Loaded PLGA Nanoparticles. Chinese Journal of Modern Applied Pharmacy 2014, 31 (6), 717-721.

Reviewer 2 Report
Comments and Suggestions for Authors
The manuscript deals with the conception of a novel hydrogel, which could be a potential candidate for applications in tissue defect repair. More specifically, a phenylboronic acid-modified oxidized hyaluronic acid was synthesized and dynamically crosslinked with catechol group-modified glycol chitosan and guar gum into a hydrogel loaded with curcumin liposomes.
Based on findings of different studies, the hydrogel possessed excellent injectable performance and tissue adaptive property with self-healing capacity, tissue adhesion, and adaptation to tissue movement. Further, the hydrogel encapsulates remarkable biocompatibility and antioxidant performances as well as more sustainable release of curcumin.
I found the experiments were well done and the manuscript is well written. However, some issues should be resolved.
Comments
1. The title should be revised to become more attractive.
2. Some key findings should be detailed and highlighted in the “Abstract” section.
3. The modification of glycol chitosan needs a reference of the previous literature.
4. The Mw of the modified glycol chitosan should be specified. As well as the DDA of the chitosan.
5. The degree of substitution of sodium hyaluronate should be specified.
6. The Mw of Guar gum should be specified.
7. The authors should specify the temperature for:
- 4. Preparation of the Hydrogels
- 5.1. Swelling ratios of the hydrogels
8. How have samples been sterilized before biological tests?
9. Figure 4C: include the symbol η* for shear viscosity.
10. Graphs in Figure 4D must be compared according to p levels. What is the significance of the difference between the results according to the p-level? In this case, the authors should use an ANOVA test.
11. Legend of Figure 4 should be revised and more detailed according to a, b, c, d, and e.
12. “Typical reaction of hydrogel” should be presented as “d” in Figure 5.
13. Protocols used for statistical analysis should be presented as a sub-section in M&M.
14. The authors should highlight the limitations and perspective of the study.
Author Response
Reviewer 2
The manuscript deals with the conception of a novel hydrogel, which could be a potential candidate for applications in tissue defect repair. More specifically, a phenylboronic acid-modified oxidized hyaluronic acid was synthesized and dynamically crosslinked with catechol group-modified glycol chitosan and guar gum into a hydrogel loaded with curcumin liposomes.
Based on findings of different studies, the hydrogel possessed excellent injectable performance and tissue adaptive property with self-healing capacity, tissue adhesion, and adaptation to tissue movement. Further, the hydrogel encapsulates remarkable biocompatibility and antioxidant performances as well as more sustainable release of curcumin.
I found the experiments were well done and the manuscript is well written. However, some issues should be resolved.
Thank you very much for improving our manuscript, the revised manuscript had been modified according to your comments. We really hope our revised manuscript could be suitable for the publication in “Polymers”.
Comments
- The title should be revised to become more attractive.
Response: Thank you very much for improving our manuscript; the title had been revised as “Sustained Release of Curcumin from Cur-LPs loaded Adaptive Injectable Self-Healing Hydrogels”.
- Some key findings should be detailed and highlighted in the “Abstract” section.
Response: Thank you very much for improving our manuscript; the Abstract had been revised to detail and highlight the key findings.
Biological tissue defects are typically characterized by various shaped defects, and prone to inflammation and excessive accumulation of reactive oxygen species. Therefore, it is still urgent to develop functional materials which can fully occupy and adhere to irregularly shaped defects by injection and promote the tissue repair process by antioxidant and anti-inflammatory. Herein, in this work phenylboronic acid modified oxidized hyaluronic acid (OHAPBA) was synthesized and dynamically crosslinked with catechol group modified glycol chitosan (GCHCA) and guar gum (GG) into a hydrogel loaded with curcumin liposomes (Cur-LPs) which were relatively uniformly distributed around 150 nm. The hydrogel possessed rapid gelation within 30 s, outstanding injectability and tissue adaptive property with self-healing property, adhesive property of biological tissues and adaptation to tissue movement. Moreover, good biocompatibility and higher DPPH scavenging efficiency were illustrated in the hydrogel. And more sustainable release of curcumin from Cur-LPs loaded hydrogels which could be last for 10 days was achieved to improve the bioavailability of curcumin. Finally, they might be injected to fully occupy and adhere to irregularly shaped defects and promote the tissue repair process by antioxidant and sustained release of curcumin for anti-inflammation. And the hydrogel would have potential application of candidates in tissue defect repair.
- The modification of glycol chitosan needs a reference of the previous literature.
Response: Thank you very much for improving our manuscript; the reference of our previous literature had been added in our revised manuscript. Catechol modified glycol chitosan (GCHCA) was prepared following the published methods in our previous literature.6
- The Mw of the modified glycol chitosan should be specified. As well as the DDA of the chitosan.
Response: Thank you very much for improving our manuscript; the Mw of the modified glycol chitosan and the DDA of the chitosan had been added in our revised manuscript. Glycol chitosan (GC, Mw 100 kDa, DDA>98%)
- The degree of substitution of sodium hyaluronate should be specified.
Response: Thank you very much for improving our manuscript; the degree of substitution of sodium hyaluronate had been specified in our revised manuscript. And the grafting rate of phenylboronic acid group was 6.8 ± 0.2% by the calculation of the integral area at 1.91 ppm corresponding to - CH3 and 7.01 ppm corresponding to the hydrogen on the benzene ring.
- The Mw of Guar gum should be specified.
Response: Thank you very much for improving our manuscript; the Mw of Guar gum had been specified in our revised manuscript. Guar gum (GG, Mw 20 kDa)
- The authors should specify the temperature for:
- 4. Preparation of the Hydrogels
- 5.1. Swelling ratios of the hydrogels
Response: Thank you very much for improving our manuscript; the temperature for Preparation of the Hydrogels and Swelling ratios of the hydrogels had been specified in our revised manuscript. Then the aforementioned GCHCA/GG solution and OHAPBA solution were mixed vigorously by vortex with equal volume ratio at room temperature to prepare the hydrogels. Swelling ratios of the hydrogels was measured in PBS (0.1 M, pH 7.4) solution at 37 ℃.
- How have samples been sterilized before biological tests?
Response: Thank you very much for improving our manuscript; the sterilization process of samples before biological tests had been added in our revised manuscript. The sterilized hydrogels were prepared in the ultraclean table by means of ultraviolet sterilization, preparation of pre-gel solution from the sterilization solution, and sterilization of the pre-gel solution through 0.22 μm filter membrane.
- Figure 4C: include the symbol η* for shear viscosity.
Response: Thank you very much for improving our manuscript; the symbol η* for shear viscosity had been added in Figure 4C of our revised manuscript.
- Graphs in Figure 4D must be compared according to p levels. What is the significance of the difference between the results according to the p-level? In this case, the authors should use an ANOVA test.
Response: Thank you very much for improving our manuscript; the Statistical differences had been added in Figure 4d of our revised manuscript.
Figure 4 a) Gelation process by inverted method; b) Injectable performance; c) Shear-thinning behavior; d) Swelling ratios; e) G′ and G″ changes versus strain
- Legend of Figure 4 should be revised and more detailed according to a, b, c, d, and e.
Response: Thank you very much for improving our manuscript; the Legend of Figure 4 had been revised to be more detailed in our revised manuscript. Figure 4 a) Gelation process by inverted method; b) Injectable performance; c) Shear-thinning behavior; d) Swelling ratios; e) G′ and G″ changes versus strain
- “Typical reaction of hydrogel” should be presented as “d” in Figure 5.
Response: Thank you very much for improving our manuscript; “typical reaction of hydrogel” had been presented as “d” in Figure 5 in our revised manuscript.
- Protocols used for statistical analysis should be presented as a sub-section in M&M.
Response: Thank you very much for improving our manuscript; the statistical analysis had been added in Section 2.7 of our revised manuscript.
- The authors should highlight the limitations and perspective of the study.
Response: Thank you very much for improving our manuscript; the limitations and perspective of the study had been highlighted in the last part of our revised manuscript.
From all the above it could be seen that curcumin liposomes were loaded into adaptive injectable self-healing hydrogels formed by a dynamic crosslinked network. The increased water solubility with efficient loading and slow release performances of curcumin indirectly proved the enhancement of bioavailability. However, the direct evidence of bioavailability is not provided in this work, which is also the limitation of this manuscript. In our future work, the direct evidence of the improvement of bioavailability and the in vivo application for tissue repair e.g. wound healing, bone regeneration or periodontitis will be investigated.
REFERENCES
- Guo, H.; Huang, S.; Yang, X.; Wu, J.; Kirk, T. B.; Xu, J.; Xu, A.; Xue, W., Injectable and Self-Healing Hydrogels with Double-Dynamic Bond Tunable Mechanical, Gel-Sol Transition and Drug Delivery Properties for Promoting Periodontium Regeneration in Periodontitis. Acs Appl Mater Inter 2021, 13 (51), 61638-61652.
- Wang, L.; Ding, X.; Fan, L.; Filppula, A. M.; Li, Q.; Zhang, H.; Zhao, Y.; Shang, L., Self-Healing Dynamic Hydrogel Microparticles with Structural Color for Wound Management. Nano-Micro Letters 2024, 16 (1), 232.
- Hua, S.; Zhang, Y.; Zhu, Y.; Fu, X.; Meng, L.; Zhao, L.; Kong, L.; Pan, S.; Che, Y., Tunicate cellulose nanocrystals strengthened injectable stretchable hydrogel as multi-responsive enhanced antibacterial wound dressing for promoting diabetic wound healing. Carbohydrate Polymers 2024, 343, 122426.
- Wu, Z.; Li, M.; Fang, H.; Wang, B., A new boronic acid based fluorescent reporter for catechol. Bioorganic & Medicinal Chemistry Letters 2012, 22 (23), 7179-7182.
- Ye, Q.; Yan, F.; Kong, D.; Zhang, J.; Zhou, X.; Xu, J.; Chen, L., Constructing a fluorescent probe for specific detection of catechol based on 4-carboxyphenylboronic acid-functionalized carbon dots. Sensors and Actuators B: Chemical 2017, 250, 712-720.
- Zhou, X.; Ning, X.; Chen, Y.; Chang, H.; Lu, D.; Pei, D.; Geng, Z.; Zeng, Z.; Guo, C.; Huang, J.; Yu, S.; Guo, H., Dual Glucose/ROS-Sensitive Injectable Adhesive Self-Healing Hydrogel with Photothermal Antibacterial Activity and Modulation of Macrophage Polarization for Infected Diabetic Wound Healing. Acs Materials Letters 2023, 5 (12), 3142-3155.
- Gao, G.; Jiang, Y.-W.; Jia, H.-R.; Wu, F.-G., Near-infrared light-controllable on-demand antibiotics release using thermo-sensitive hydrogel-based drug reservoir for combating bacterial infection. Biomaterials 2019, 188, 83-95.
- Tang, P.; Han, L.; Li, P.; Jia, Z.; Wang, K.; Zhang, H.; Tan, H.; Guo, T.; Lu, X., Mussel-Inspired Electroactive and Antioxidative Scaffolds with Incorporation of Polydopamine-Reduced Graphene Oxide for Enhancing Skin Wound Healing. Acs Appl Mater Inter 2019, 11 (8), 7703-7714.
- Feng, X.; Pi, C.; Fu, S.; Yang, H.; Zheng, X.; Hou, Y.; Wang, Y.; Zhang, X.; Zhao, L.; Wei, Y., Combination of Curcumin and Paclitaxel Liposomes Exhibits Enhanced Cytotoxicity Towards A549/A549-T Cells and Unaltered Pharmacokinetics. Journal of biomedical nanotechnology 2020, 16 (8), 1304-1313.
- Li, Z.-l.; Peng, S.-f.; Chen, X.; Zhu, Y.-q.; Zou, L.-q.; Liu, W.; Liu, C.-m., Pluronics modified liposomes for curcumin encapsulation: Sustained release, stability and bioaccessibility. Food Research International 2018, 108, 246-253.
- Sun, D.; Zhou, J.-K.; Zhao, L.; Zheng, Z.-Y.; Li, J.; Pu, W.; Liu, S.; Liu, X.-S.; Liu, S.-J.; Zheng, Y.; Zhao, Y.; Peng, Y., Novel Curcumin Liposome Modified with Hyaluronan Targeting CD44 Plays an Anti-Leukemic Role in Acute Myeloid Leukemia <i>in Vitro</i> and <i>in Vivo</i>. Acs Appl Mater Inter 2017, 9 (20), 16858-16869.
- Mimica, B.; Popovic, V. B.; Banjari, I.; Kadic, A. J.; Puljak, L., Methods Used for Enhancing the Bioavailability of Oral Curcumin in Randomized Controlled Trials: A Meta-Research Study. Pharmaceuticals 2022, 15 (8), 939.
- Shoji, M.; Nakagawa, K.; Watanabe, A.; Tsuduki, T.; Yamada, T.; Kuwahara, S.; Kimura, F.; Miyazawa, T., Comparison of the effects of curcumin and curcumin glucuronide in human hepatocellular carcinoma HepG2 cells. Food Chemistry 2014, 151, 126-132.
- Alavi, F.; Ciftci, O. N., Increasing the bioavailability of curcumin using a green supercritical fluid technology-assisted approach based on simultaneous starch aerogel formation-curcumin impregnation. Food Chemistry 2024, 455, 139468.
- Zheng, Y.; Zheng, S.; Wang, Z., Enhancement of Oral Bioavailability of Curcumin Loaded PLGA Nanoparticles. Chinese Journal of Modern Applied Pharmacy 2014, 31 (6), 717-721.

Reviewer 3 Report
Comments and Suggestions for Authors
The manuscript is a significant contribution to the field of hydrogels, their making, and their application to tissue repair. The authors show the sustainable release of curcumin from Cur-LPs-loaded hydrogels, which might improve curcumin's bioavailability to enhance the hydrogels' anti-inflammation performance. The manuscript is well-written and summarized. This reviewer has some suggestions.
1. The authors showed the formation of Cur-LPs, and it is assumed that the modification improved the bioavailability of curcumin. What is the basis of this hypothesis? If possible, please provide some evidence in support of this hypothesis.
2. There are inconsistencies in paragraph 2.2. there should be a space between numbers and units; please correct this throughout the manuscript.
3. Picture quality in figure 4d and 4e are not good. Please provide high-resolution pictures.
4. There are some inconsistencies in the font size; please pay some concentration. On page 9, line 256, the font size differs from the other. A few other lines are also in different formats.
5. A few references are not in the correct format, e.g., reference 11 is given in the wrong format. Please pay attention to correct these.
6. Some sentences are somewhat difficult to understand. Please pay some attention to correct those sentences. Page 8, line 231.

Author Response
Reviewer 3
The manuscript is a significant contribution to the field of hydrogels, their making, and their application to tissue repair. The authors show the sustainable release of curcumin from Cur-LPs-loaded hydrogels, which might improve curcumin's bioavailability to enhance the hydrogels' anti-inflammation performance. The manuscript is well-written and summarized. This reviewer has some suggestions.
Thank you very much for improving our manuscript, the revised manuscript had been modified according to your comments. We really hope our revised manuscript could be suitable for the publication in “Polymers”.
- The authors showed the formation of Cur-LPs, and it is assumed that the modification improved the bioavailability of curcumin. What is the basis of this hypothesis? If possible, please provide some evidence in support of this hypothesis.
Response: Thank you very much for improving our manuscript; Curcumin is a poorly soluble drug with a slow dissolution rate, resulting in poor absorption and low bioavailability in the body, which limits its clinical application. 9-11 It was reported that a drug delivery system which can increase water solubility with efficient loading and slow release can enhance the bioavailability of curcumin. 12-15 And in our work, the curcumin content in liposomes, calculated by the standard curve of curcumin measured by UV-vis spectroscopy (Figure 2d), was 97 μg/mg with 64% encapsulation efficiency. In order to further assess the application performance of Cur-LPs loaded OHAPBA/GCHCA/GG hydrogel in biomedicine, in vitro curcumin (an anti-inflammatory agent) release from the liposome and the hydrogel was analyzed, as was shown in Figure 6d. More sustained curcumin release performance illustrated in the Cur-LPs loaded OHAPBA/GCHCA/GG hydrogel, comparing that release from liposomes. The more sustained curcumin release property could extend the action time, reduce the frequency of administration, and finally improve the bioavailability of curcumin. The increased water solubility with efficient loading and slow release performances of curcumin indirectly proved the enhancement of bioavailability. However, the direct evidence of bioavailability is not provided in this manuscript, which is also the limitation of this manuscript. In our future work, the direct evidence of the improvement of bioavailability and the in vivo application for tissue repair e.g. wound healing, bone regeneration or periodontitis will be investigated. And the limitation of this manuscript had been supplied at the end of our revised manuscript. We really hope the reviewer could agree with this part we modified in our revised manuscript. Thank you, again.
- There are inconsistencies in paragraph 2.2. there should be a space between numbers and units; please correct this throughout the manuscript.
Response: Thank you very much for improving our manuscript; the space between numbers and units had been unified throughout our revised manuscript.
- Picture quality in figure 4d and 4e are not good. Please provide high-resolution pictures.
Response: Thank you very much for improving our manuscript; high-resolution pictures of figure 4d and 4e had been added in our revised manuscript.
Figure 4 a) Gelation process by inverted method; b) Injectable performance; c) Shear-thinning behavior; d) Swelling ratios; e) G′ and G″ changes versus strain
- There are some inconsistencies in the font size; please pay some concentration. On page 9, line 256, the font size differs from the other. A few other lines are also in different formats.
Response: Thank you very much for improving our manuscript; the font format had been unified throughout our revised manuscript.
- A few references are not in the correct format, e.g., reference 11 is given in the wrong format. Please pay attention to correct these.
Response: Thank you very much for improving our manuscript; all the references had been checked and corrected throughout our revised manuscript.
- Some sentences are somewhat difficult to understand. Please pay some attention to correct those sentences. Page 8, line 231.
Response: Thank you very much for improving our manuscript; Page 8, line 231. had been corrected as “As was shown in Figure 5a, when the two halves of the cracked hydrogels were contacted immediately, they would be healed to an integrated hydrogel, which was capable of withstanding certain tension without damaging.” in our revised manuscript, and also all the sentences had been checked and corrected throughout our revised manuscript.
REFERENCES
- Guo, H.; Huang, S.; Yang, X.; Wu, J.; Kirk, T. B.; Xu, J.; Xu, A.; Xue, W., Injectable and Self-Healing Hydrogels with Double-Dynamic Bond Tunable Mechanical, Gel-Sol Transition and Drug Delivery Properties for Promoting Periodontium Regeneration in Periodontitis. Acs Appl Mater Inter 2021, 13 (51), 61638-61652.
- Wang, L.; Ding, X.; Fan, L.; Filppula, A. M.; Li, Q.; Zhang, H.; Zhao, Y.; Shang, L., Self-Healing Dynamic Hydrogel Microparticles with Structural Color for Wound Management. Nano-Micro Letters 2024, 16 (1), 232.
- Hua, S.; Zhang, Y.; Zhu, Y.; Fu, X.; Meng, L.; Zhao, L.; Kong, L.; Pan, S.; Che, Y., Tunicate cellulose nanocrystals strengthened injectable stretchable hydrogel as multi-responsive enhanced antibacterial wound dressing for promoting diabetic wound healing. Carbohydrate Polymers 2024, 343, 122426.
- Wu, Z.; Li, M.; Fang, H.; Wang, B., A new boronic acid based fluorescent reporter for catechol. Bioorganic & Medicinal Chemistry Letters 2012, 22 (23), 7179-7182.
- Ye, Q.; Yan, F.; Kong, D.; Zhang, J.; Zhou, X.; Xu, J.; Chen, L., Constructing a fluorescent probe for specific detection of catechol based on 4-carboxyphenylboronic acid-functionalized carbon dots. Sensors and Actuators B: Chemical 2017, 250, 712-720.
- Zhou, X.; Ning, X.; Chen, Y.; Chang, H.; Lu, D.; Pei, D.; Geng, Z.; Zeng, Z.; Guo, C.; Huang, J.; Yu, S.; Guo, H., Dual Glucose/ROS-Sensitive Injectable Adhesive Self-Healing Hydrogel with Photothermal Antibacterial Activity and Modulation of Macrophage Polarization for Infected Diabetic Wound Healing. Acs Materials Letters 2023, 5 (12), 3142-3155.
- Gao, G.; Jiang, Y.-W.; Jia, H.-R.; Wu, F.-G., Near-infrared light-controllable on-demand antibiotics release using thermo-sensitive hydrogel-based drug reservoir for combating bacterial infection. Biomaterials 2019, 188, 83-95.
- Tang, P.; Han, L.; Li, P.; Jia, Z.; Wang, K.; Zhang, H.; Tan, H.; Guo, T.; Lu, X., Mussel-Inspired Electroactive and Antioxidative Scaffolds with Incorporation of Polydopamine-Reduced Graphene Oxide for Enhancing Skin Wound Healing. Acs Appl Mater Inter 2019, 11 (8), 7703-7714.
- Feng, X.; Pi, C.; Fu, S.; Yang, H.; Zheng, X.; Hou, Y.; Wang, Y.; Zhang, X.; Zhao, L.; Wei, Y., Combination of Curcumin and Paclitaxel Liposomes Exhibits Enhanced Cytotoxicity Towards A549/A549-T Cells and Unaltered Pharmacokinetics. Journal of biomedical nanotechnology 2020, 16 (8), 1304-1313.
- Li, Z.-l.; Peng, S.-f.; Chen, X.; Zhu, Y.-q.; Zou, L.-q.; Liu, W.; Liu, C.-m., Pluronics modified liposomes for curcumin encapsulation: Sustained release, stability and bioaccessibility. Food Research International 2018, 108, 246-253.
- Sun, D.; Zhou, J.-K.; Zhao, L.; Zheng, Z.-Y.; Li, J.; Pu, W.; Liu, S.; Liu, X.-S.; Liu, S.-J.; Zheng, Y.; Zhao, Y.; Peng, Y., Novel Curcumin Liposome Modified with Hyaluronan Targeting CD44 Plays an Anti-Leukemic Role in Acute Myeloid Leukemia <i>in Vitro</i> and <i>in Vivo</i>. Acs Appl Mater Inter 2017, 9 (20), 16858-16869.
- Mimica, B.; Popovic, V. B.; Banjari, I.; Kadic, A. J.; Puljak, L., Methods Used for Enhancing the Bioavailability of Oral Curcumin in Randomized Controlled Trials: A Meta-Research Study. Pharmaceuticals 2022, 15 (8), 939.
- Shoji, M.; Nakagawa, K.; Watanabe, A.; Tsuduki, T.; Yamada, T.; Kuwahara, S.; Kimura, F.; Miyazawa, T., Comparison of the effects of curcumin and curcumin glucuronide in human hepatocellular carcinoma HepG2 cells. Food Chemistry 2014, 151, 126-132.
- Alavi, F.; Ciftci, O. N., Increasing the bioavailability of curcumin using a green supercritical fluid technology-assisted approach based on simultaneous starch aerogel formation-curcumin impregnation. Food Chemistry 2024, 455, 139468.
- Zheng, Y.; Zheng, S.; Wang, Z., Enhancement of Oral Bioavailability of Curcumin Loaded PLGA Nanoparticles. Chinese Journal of Modern Applied Pharmacy 2014, 31 (6), 717-721.

Round 2
Reviewer 1 Report
Comments and Suggestions for Authors
1) Introduction. Lines 41-47
“Phenylboronester bond, imine bond, acylhydrazone bond, coordination complex bond, Diels-Alder reaction, or disulfide bond, such of these dynamical bonds can lead to a dynamic crosslinked network in the hydrogels, resulting in an injectablilty and self-healing property of the hydrogels, which may finally enable the hydrogels to be capable of being injected to fully occupy the tissue defects with irregular shapes, and also can be adapted to the movement of tissues and the forces from external in a completely integrated hydrogel state”
Authors for the first time here state the terms “dynamical” bonds and “dynamic” crosslinked network, which they use along the all manuscript.
Chemical bons can be classified as strong chemical bonds (intramolecular forces), such as covalent, ionic, metallic bonds and weak chemical bonds (intermolecular forces), such as Van der Waals forces, Keesom dipoles forces, London dispersion forces, hydrogen bonds.
Crosslinked networks (gels) can be chemically crosslinked (e.g., by covalent bonds) or physically crosslinked (e.g., by ionic bonds).
These are commonly accepted terms, which are widely used for description of such systems.
When speaking about rather new classes of macromolecular systems, such as vitrimers, covalent adaptable networks (CANs) – systems which can be broken and reformed by an external stimulus, authors should give a brief explanation of this technology, mention that these “dynamic” bonds are simply reversible covalent bonds, and clarify what this type of bonding gives to them and prepared gel systems. Also please clarify which type of networks you have in this work: dissociative CANs or associative CANs.
So, when you are writing about rather new type of material it is better to clarify how this group is referred to generally termed bonds, materials and networks, to make the article more easily perceived by readers.
2) Results and Discussion. Line 221.
Authors claim PDI=1.00 for liposome hydrodynamic size determination.
Also, in experimental section it is mentioned that hydrodynamic size was determined by DLS (Malvern 125 Zetasizer Nano ZS90, UK).
The polydispersity index (PDI) for Zetasizer Nano is a number calculated from a simple two parameter fit to the correlation data called a cumulants analysis. The maximum value is arbitrarily limited to 1.0.
A PDI value of 1 indicates that the sample has a very broad size distribution and may contain large particles or aggregates that could be slowly sedimenting, but authors write that “average particle size in 150 nm with narrow distribution of hydrodynamic size” (lines 220-221).
3) Manuscript still contains some typos, which should be corrected (“injectablilty” line 47 and so on).
Author Response
Reviewer 1
1) Introduction. Lines 41-47
“Phenylboronester bond, imine bond, acylhydrazone bond, coordination complex bond, Diels-Alder reaction, or disulfide bond, such of these dynamical bonds can lead to a dynamic crosslinked network in the hydrogels, resulting in an injectablilty and self-healing property of the hydrogels, which may finally enable the hydrogels to be capable of being injected to fully occupy the tissue defects with irregular shapes, and also can be adapted to the movement of tissues and the forces from external in a completely integrated hydrogel state”
Authors for the first time here state the terms “dynamical” bonds and “dynamic” crosslinked network, which they use along the all manuscript.
Chemical bons can be classified as strong chemical bonds (intramolecular forces), such as covalent, ionic, metallic bonds and weak chemical bonds (intermolecular forces), such as Van der Waals forces, Keesom dipoles forces, London dispersion forces, hydrogen bonds.
Crosslinked networks (gels) can be chemically crosslinked (e.g., by covalent bonds) or physically crosslinked (e.g., by ionic bonds).
These are commonly accepted terms, which are widely used for description of such systems.
When speaking about rather new classes of macromolecular systems, such as vitrimers, covalent adaptable networks (CANs) – systems which can be broken and reformed by an external stimulus, authors should give a brief explanation of this technology, mention that these “dynamic” bonds are simply reversible covalent bonds, and clarify what this type of bonding gives to them and prepared gel systems. Also please clarify which type of networks you have in this work: dissociative CANs or associative CANs.
So, when you are writing about rather new type of material it is better to clarify how this group is referred to generally termed bonds, materials and networks, to make the article more easily perceived by readers.
Response: Thank you very much for improving our manuscript; we had revised this part according to your useful requirement. Dynamical bonds which are reversible covalent bonds, dynamic crosslinked networks which are reversible covalently crosslinked network or covalent adaptable networks, and this type of bonding that can result in reversible performances such as shear-thinning (injectability) and self-healing property of the hydrogels had been added in our revised manuscript. Also the potential dissociative reversion mechanisms 1 of covalent adaptable networks dynamic crosslinked by dynamic phenylboronic esters bonds were illustrated schematically in Figure 5c.
Phenylboronester bond, imine bond, acylhydrazone bond, coordination complex bond, Diels-Alder reaction, or disulfide bond, such of these dynamical bonds which are reversible covalent bonds can lead to dynamic crosslinked networks which are reversible covalently crosslinked network or covalent adaptable networks2-3 in the hydrogels, resulting in reversible performances such as shear-thinning (injectability) and self-healing property of the hydrogels
The hydrogel crosslinked through dynamic chemical bonds could be injected through a needle of 26G easily without blocking the needle, as was shown in Figure 4b. Stable self-healing behavior illustrated in Figure 5b after four alternative strain cycles, which was attributed to the hydrogel network crosslinked by dynamic bonds. And the potential dissociative reversion mechanisms 1 of covalent adaptable networks dynamic crosslinked by dynamic phenylboronic esters bonds were illustrated schematically in Figure 5c.
2) Results and Discussion. Line 221.
Authors claim PDI=1.00 for liposome hydrodynamic size determination.
Also, in experimental section it is mentioned that hydrodynamic size was determined by DLS (Malvern 125 Zetasizer Nano ZS90, UK).
The polydispersity index (PDI) for Zetasizer Nano is a number calculated from a simple two parameter fit to the correlation data called a cumulants analysis. The maximum value is arbitrarily limited to 1.0.
A PDI value of 1 indicates that the sample has a very broad size distribution and may contain large particles or aggregates that could be slowly sedimenting, but authors write that “average particle size in 150 nm with narrow distribution of hydrodynamic size” (lines 220-221).
Response: Thank you very much for pointing out this problem; We retested DLS with other instrument (Zetasizer Nano ZS, Malvern, UK), and the results were illustrated in Figure 2a. From Figure 2a it could be seen that average particle size in 180 nm with relatively narrow distribution of hydrodynamic size (PDI=0.45) illustrate in the DLS results.
3) Manuscript still contains some typos, which should be corrected (“injectablilty” line 47 and so on).
Response: Thank you very much for improving our manuscript; the “injectablilty” had been corrected as “injectability”.
REFERENCES
- Bapat, A. P.; Roy, D.; Ray, J. G.; Savin, D. A.; Sumerlin, B. S., Dynamic-Covalent Macromolecular Stars with Boronic Ester Linkages. Journal of the American Chemical Society 2011, 133 (49), 19832-19838.
- Chang, L.; Wang, C.; Han, S.; Sun, X.; Xu, F., Chemically Triggered Hydrogel Transformations through Covalent Adaptable Networks and Applications in Cell Culture. ACS Macro Letters 2021, 10 (7), 901-906.
- Podgórski, M.; Fairbanks, B. D.; Kirkpatrick, B. E.; McBride, M.; Martinez, A.; Dobson, A.; Bongiardina, N. J.; Bowman, C. N., Toward Stimuli-Responsive Dynamic Thermosets through Continuous Development and Improvements in Covalent Adaptable Networks (CANs). Adv Mater 2020, 32 (20), 1906876.
Reviewer 2 Report
Comments and Suggestions for Authors
The authors addressed some concerns. However, these minors should be considered:
1. The authors should specify how these specifications have been obtained. Determined by the authors? or specified by the suppliers?
· Glycol chitosan (GC, Mw 100 kDa, DDA>98%)
· Guar gum (GG, Mw 20 kDa)
This information should be specified in the final revised version of the manuscript (M&M section).
2. Following my previous comment The authors highlight in the revised manuscript the grafting rate of the phenylboronic acid-modified oxidized hyaluronic acid. This is good; however, my previous comment concerns the degree of substitution of the raw (unmodified) sodium hyaluronate, which should be specified.
This information should be specified in the final revised version of the manuscript.
3. For the UV sterilization, the authors should identify the device of the UV light source used for that. As well as the concertation of exposition (mW/cm2) and exposition time.
This information should be specified in the final revised version of the manuscript.
4. In Figure 4 C, the shear viscosity η* should be written as Complex viscosity η*. Further, the term “shear viscosity” should be replaced with “complex viscosity” throughout the manuscript.
5. The authors should specify in section 2.7 (Statistical analysis) how many samples or how many measurements have been carried out per condition in this study.
Author Response
Reviewer 2
The authors addressed some concerns. However, these minors should be considered:
Thank you very much for improving our manuscript, the minor concerns had been modified according to your comments. We really hope our revised manuscript could be suitable for the publication in “Polymers”.
- The authors should specify how these specifications have been obtained. Determined by the authors? or specified by the suppliers?
- Glycol chitosan (GC, Mw 100 kDa, DDA>98%)
- Guar gum (GG, Mw 20 kDa)
This information should be specified in the final revised version of the manuscript (M&M section).
Response: Thank you very much for improving our manuscript; the information had been specified in the final revised version of the manuscript (M&M section). The molecular weight or other indicators of these raw materials are specified by the suppliers.
- Following my previous comment The authors highlight in the revised manuscript the grafting rate of the phenylboronic acid-modified oxidized hyaluronic acid. This is good; however, my previous comment concerns the degree of substitution of the raw (unmodified) sodium hyaluronate, which should be specified.
This information should be specified in the final revised version of the manuscript.
Response: Thank you very much for improving our manuscript; the information (Sodium Hyaluronate without substitution (HA, 1000~1500 kDa)) had been specified in the final revised version of the manuscript (M&M section).
- For the UV sterilization, the authors should identify the device of the UV light source used for that. As well as the concertation of exposition (mW/cm2) and exposition time.
This information should be specified in the final revised version of the manuscript.
Response: Thank you very much for improving our manuscript; the information had been specified in the final revised version of the manuscript (M&M section). UV sterilization were performed in the ultraclean table (BBS-DDC, BIOBASE, China) with 3 mW/cm2 for 30 min.
- In Figure 4 C, the shear viscosity η* should be written as Complex viscosity η*. Further, the term “shear viscosity” should be replaced with “complex viscosity” throughout the manuscript.
Response: Thank you very much for improving our manuscript; In Figure 4 C, the shear viscosity η* had been written as Complex viscosity η*. And the term “shear viscosity” had been replaced with “complex viscosity” throughout the revised manuscript.
- The authors should specify in section 2.7 (Statistical analysis) how many samples or how many measurements have been carried out per condition in this study.
Response: Thank you very much for improving our manuscript; the number of samples or measurements had been added in section 2.7 (Statistical analysis) of the final revised version of the manuscript. Every group with five parallel data was presented as mean ± SD.